# FaceDNeRF: Semantics-Driven Face Reconstruction, Prompt Editing and Relighting with Diffusion Models

Hao Zhang*
HKUST
hzhangcc@connect.ust.hk

Yanbo Xu*
CMU, HKUST
yxubu@connect.ust.hk

Tianyuan Dai*
Stanford University, HKUST
tdaiaa@connect.ust.hk

Yu-Wing Tai
Dartmouth College
yuwing@gmail.com

Chi-Keung Tang
HKUST
cktang@cs.ust.hk

## Abstract

The ability to create high-quality 3D faces from a single image has become increasingly important with wide applications in video conferencing, AR/VR, and advanced video editing in movie industries. In this paper, we propose Face Diffusion NeRF (FaceDNeRF), a new generative method to reconstruct high-quality Face NeRFs from single images, complete with semantic editing and relighting capabilities. FaceDNeRF utilizes high-resolution 3D GAN inversion and expertly trained 2D latent-diffusion model, allowing users to manipulate and construct Face NeRFs in zero-shot learning without the need for explicit 3D data. With carefully designed illumination and identity preserving loss, as well as multi-modal pre-training, FaceDNeRF offers users unparalleled control over the editing process enabling them to create and edit face NeRFs using just single-view images, text prompts, and explicit target lighting. The advanced features of FaceDNeRF have been designed to produce more impressive results than existing 2D editing approaches that rely on 2D segmentation maps for editable attributes. Experiments show that our FaceDNeRF achieves exceptionally realistic results and unprecedented flexibility in editing compared with state-of-the-art 3D face reconstruction and editing methods. Our code will be available at `https://github.com/BillyXYB/FaceDNeRF`.

## 1   Introduction

Rich and versatile 3D contents are in high demand in entertainment industries such as movie making, computer gaming and emerging applications such as Metaverse, which also have high potential in robotics learning as well, where high-quality synthetic data in large amounts are required to enhance generalizability. 3D generative methods such as EG3D [8] can generate high-fidelity NeRF from a single image, but the reconstructed NeRF cannot be easily controlled or edited. Some methods [33, 57] use pixel-wise segmentation maps or user-supplied sketches to guide 3D editing. But these methods are hard to scale up due to the demanding editing requirement and limited editable attributes. Language is regarded as one of the most suitable candidates to provide control signals for 3D editing, especially given the current success of 2D semantics-driven editing [28, 42], where image and language domains are bridged by CLIP, i.e., Contrastive Language-Image Pre-training [45]. However, there is still substantial room for improvement in 3D generative and editable models, in terms of usability, edit-ability, and results quality.

---

*These authors contributed equally to this work.

37th Conference on Neural Information Processing Systems (NeurIPS 2023).

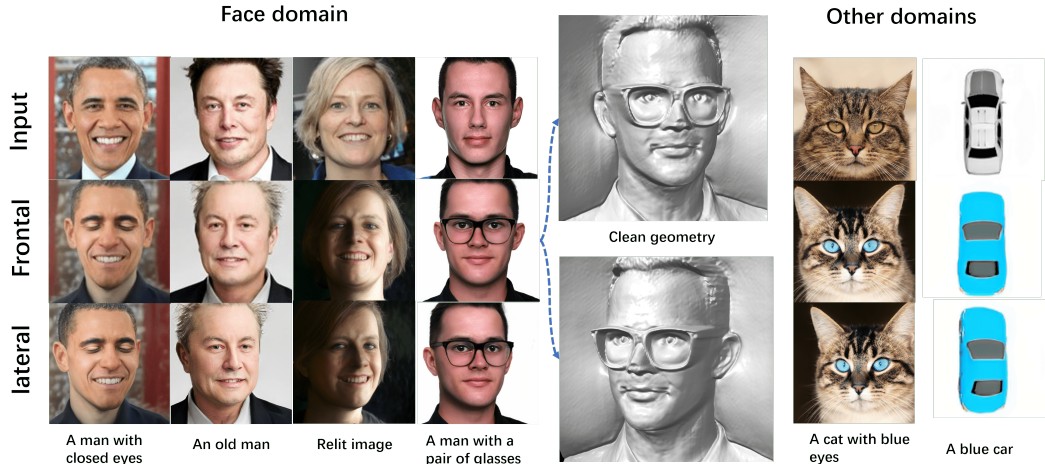

Figure 1: **FaceDNeRF results.** Given a face image, FaceDNeRF can reconstruct, edit, and relit a photo-realistic 3D face NeRF using a text prompt and a target light. Our method is not limited to human faces, and can be applied to other domains as well. Further results and videos are available in the supplemental materials.

Neural Radiance Field (NeRF), a new paradigm in 3D scene representation, embeds a 3D scene in a compact fully-connected neural network and achieves realistic rendering of novel views. NeRF is optimized to approximate a continuous and thus differentiable scene representation function, which has quickly become a dominant approach for many relevant downstream tasks including novel scene composition [29, 38, 41, 66, 70], surface reconstruction [31, 61, 68], and articulated 3D shape reconstruction [10, 23, 43, 63, 67, 71] due to its differentiability, multi-view consistency, compactness, and fast inference. Since NeRF relies on training from multi-view images, its relighting is contingent upon the relighting of these 2D images. Nevertheless, current 2D relighting techniques [5, 22, 34, 15] are unable to ensure view consistency among the relit multi-view images, resulting in a decline in the quality of the relit NeRF.

Significant attempts have been made in recent years to develop semantic-driven NeRF editing models. Pre-trained 2D language-image models are usually leveraged to enable multi-modality in 3D. CLIP-NeRF [60] introduces a disentangled conditional NeRF architecture, where the disentangled latent representations can be bridged by two mappers trained with a CLIP-based matching loss to the CLIP embedding, based on which the latent codes are updated for editing based on the input text prompt. Outperforming CLIP-based methods, DreamFusion [44] later adopts Imagen [49], a pre-trained text-to-image diffusion model, as a prior to guide the optimization of parameters of a randomly initialized NeRF through a novel SDS loss. Magic3D [32] further improves DreamFusion by introducing a coarse-to-fine optimization scheme with diffusion priors at different resolutions. However, these methods have two main limitations: lack of photorealism and long inference time (reportedly taking 1.5 hours for DreamFusion, and 40 minutes for Magic3D).

We believe the main reason underlying the less realistic results and slow inference of DreamFusion [44] and its variants lies in the random and thus unrestricted initialization of NeRF. Leveraging 3D generative models such as EG3D [8], whose generation is restricted by the discriminator during training for constrained NeRF generation, may offer a feasible solution. Although combining 2D generative models with CLIP [45] for realistic image generation and editing has been a widely adopted approach [55], no significant attempts have been made in 3D realistic context to our knowledge. Thus, in this paper, we present FaceDNeRF, which is to our knowledge the first work to enable semantic-driven NeRF editing in 3D with high photorealism and versatile relighting from a single image, given a text prompt and target light. FaceDNeRF freezes the weights of the EG3D network which can reconstruct photorealistic NeRF from a single image. To enable semantic-driven editing, we adopt stable diffusion [48] to guide the optimization through SDS loss introduced in DreamFusion [44]. Based on SDS loss, identity loss, and feature loss, the latent vector in EG3D's latent space can be updated. Moreover, the proposed illumination loss allows explicit control over the lighting in a view-consistent manner.

## 2 Related Work

**2D Generation and Editing**    Research on unconditional or text-driven image generation has made fruitful progress. Generative Adversarial Networks (GANs) [17] contribute to the first revolutionary 2D image generative methods, among which StyleGAN [26] and its variants [24, 27] stand out due to the expressive and well-disentangled latent spaces. StyleGAN-based image editing requires either an encoder trained to map a given image to the latent space [2, 39], or specifying latent update direction which requires explicit ground-truth annotations [1, 19, 59, 65]. Diffusion models [54, 56] represent another class of generative models that enables text-driven, photorealistic and highly diverse image generation. State-of-the-art text-to-image synthesis methods contribute effective mechanisms to guide samples toward semantics: classifier-free guidance [20, 37] generates images with or without class information during model training; CLIP guidance [13, 42, 69] where CLIP [45] trained on 400 million image-text pairs spearheaded cross-modal representation learning in modern vision-language tasks. Leveraging the rich joint embedding spaces of CLIP and expressiveness of diffusion models, stable diffusion [48] is arguably the best text-to-image model to date, synthesizing images of vast domains and styles based on a text prompt. Thus we adopt stable diffusion as the guidance model in our 3D approach.

**3D Generation and Editing**    NeRF [36], an implicit neural representation, has become the dominating modern approach for 3D generation due to its continuity, differentiability, compactness, and quality of novel-view synthesis over mesh and point cloud. GRAF [50] combines implicit neural rendering with GAN for generalizable NeRF. PiGAN [7] utilizes SiREN [53] to condition the implicit neural radiance field on the latent space. Although guaranteed with 3D consistency, volumetric rendering requires heavy computation. With limited computation, the image quality of these methods is still not comparable to those produced by current state-of-the-art 2D GANs. Thus, many recent approaches adopt hybrid structures. StyleNeRF [18] applies volume rendering in the early feature maps in low resolution, followed by upsampling blocks to generate high-resolution images. However, a regularizer based on NeRF is required to ensure 3D consistency during upsampling. Instead of using volume rendering in early layers, EG3D [8] performs the operation on a relatively high-resolution feature map using a hybrid representation for 3D features generated by StyleGAN2 [27] backbone, named tri-plane, which is capable of incorporating more information than an explicit structure such as voxel. StyleSDF [40] shares a similar spirit but uses SiREN [53] for its mapping network, with the mapped result used as the input feature map followed by a style-based generator for upsampling.

Attempts have been made to generate 3D objects using diffusion models. Rodin [62], Realfusion [35], Set-the-Scene [12] and other diffusion-based 3D reconstruction methods have demonstrated the feasibility of constructing a face or other object from a single view image. However, the results are not adequately realistic with limited generalizability due to the scarcity of 3D data. It is also possible to approximate 3D assets using trained 2D diffusion models, as shown in [44, 32]. However, the quality of generated assets is visually worse than the outputs from the utilized 2D models, especially in areas such as the face, where high fidelity is an essential criterion.

**Illumination Control**    3D editable lighting on NeRF is a highly desirable feature. Image relighting methods can be roughly categorized into two groups: one involves estimating 3D face information such as 3DMM coefficients [5], albedo, and surface normals, and combining this information with a target lighting condition represented by spherical harmonics (SH) coefficients to generate relit images, such as [22, 34, 15]. Although guaranteed with 3D consistency, these methods have limited editability as they cannot easily accommodate changes in the original model, such as wearing glasses or different hairstyles. The other approach involves leveraging 2D/3D generative models to control illumination in the latent space, such as [4, 30]. While this approach is conducive to some editability, illumination can only be implicitly controlled by latent codes, that is, environmental lighting cannot be directly controlled by e.g., SH coefficients or cube mapping. Our method is amenable to both light editing means, bypassing any intrinsic separation which is not necessary in FaceDNeRF.

## 3 Method

### 3.1 EG3D and $\mathcal{W}^+$ Space

Our method utilizes trained EG3D generator [8] with its respective latent space. From initial latent code $z \in \mathcal{R}^{512}$, a mapping network $\mathcal{M}$ maps $z$ to $w$ in the space named $\mathcal{W}$, where $w \in \mathcal{R}^{1 \times 512}$.

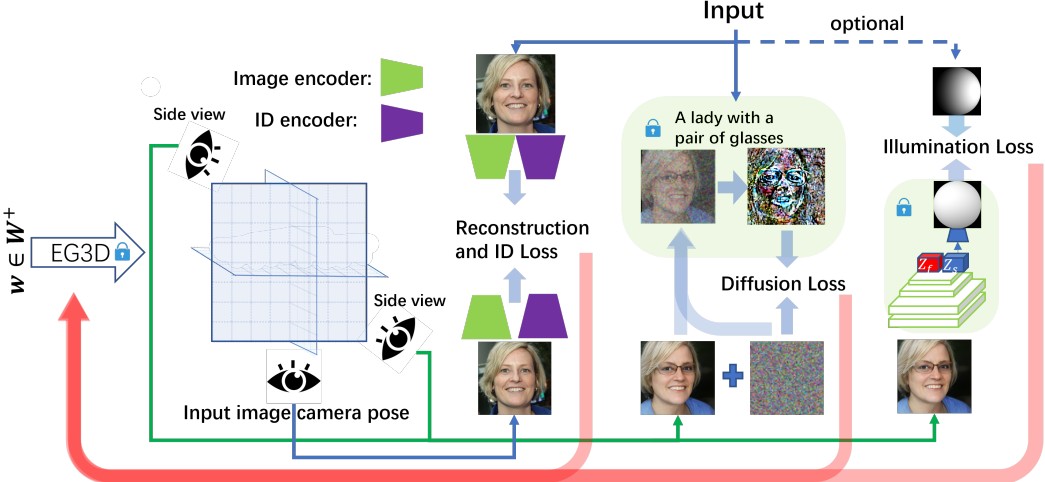

Figure 2: **FaceDNeRF structure**. The Latent 512 scalars are initialized as the mean of sampled $\mathcal{W}^+$ codes. The model computes the Reconstruction Loss and Identity Loss from the input image's camera view. From other side views, given a text prompt, the model computes the Diffusion Loss by assessing the discrepancy between the predicted noise and random noise. The Illumination Loss is then computed by comparing the estimated SH coefficients of the rendered side-view image to the target SH coefficients. These Losses are then utilized to update the Latent 512 scalars iteratively through a carefully designed differentiable model via back-propagation.

During training and generation, the $w$ code will be utilized to modulate all convolution layers as in StyleGAN2 [27]. The generated features will be reshaped into three orthogonal feature plans $(F_{xy}, F_{yz}, F_{xz})$, where each of them has the resolution of $N \times N \times C$. Given a camera pose, an augmented feature can be rendered using volume rendering as in [36], which will then be up-sampled by the super-resolution blocks. The rendering process acts as an inductive bias that enforces view-consistent results.

The inversion process inverses an input image to the latent space, such that the inverted code $w'$ can faithfully reconstruct the given input. As shown in 2D GAN inversion [46], the quality of reconstruction is better when the inversion is conducted on $\mathcal{W}^+$ space, where its latent codes $w^+ \in \mathcal{R}^{L \times 512}$ are used to separately modulate all $L$ convolution blocks. In addition, the $w^+$ space preserves the editability [64, 58]. Thus, the reconstruction and editing process of our method operates on the $\mathcal{W}^+$ space.

### 3.2 Formulation

Fig. 2 summarizes our method, which takes a single image $x$, text prompt $y$, and target lighting represented by SH coefficients $l$ as inputs, and generates a reconstructed 3D face NeRF as output. Inspired by high-fidelity face NeRF reconstruction techniques, such as EG3D inversion [8]. The Reconstruction and Identity Preservation (Sec. 3.3) is employed to guide the facial NeRF reconstruction process. To enable language-guided editing, we incorporate a Diffusion component (Sec. 3.4), which is based on the trained text-conditioned latent diffusion model. Notably, our method allows for explicit illumination control, thanks to the inclusion of the Illumination component (Sec. 3.5). To generate the target latent code, we randomly sample a value near the mean value of the latent space as a starting point. Then we solve an optimization problem to obtain the desired output.

$\mathcal{L}_R, \mathcal{L}_{ID}, \mathcal{L}_D$ and $\mathcal{L}_{IL}$ denote the Reconstruction loss, Identity Loss, Diffusion Loss and Illumination Loss, which will be elaborated in Sec. 3.3, Sec. 3.4 and Sec. 3.5. We solve the following optimization:

$$\underset{w \in \mathcal{W}+}{\arg\min} \;\; \lambda_{ID}\mathcal{L}_{ID}(G(w,c),x) + \lambda_R\mathcal{L}_R(G(w,c),x) + \lambda_D\mathcal{L}_D(G(w,c_s),y) + \lambda_{IL}\mathcal{L}_{IL}(G(w,c_s),l_{c_s}).$$

$$(1)$$

where the $G(\cdot, \cdot)$ is the EG3D generator, $c$ is the camera pose of the input image (which can be estimated by [16]), $c_s$ is the random side camera pose. $l_{c_s}$ is the target illumination which varies with $c_s$. $\lambda_{ID}$, $\lambda_R$, $\lambda_D$ and $\lambda_{IL}$ are weights of the corresponding loss functions. We optimize the $w$ iteratively through gradient descent by back-propagating the gradient of the objective Eq. (1) through these four weighted differentiable loss functions.

### 3.3   Reconstruction and Identity Preservation

The desired editing should preserve the background and identity of the input image, and hence we design the Reconstruction Loss and Identity Loss. Given input image $x$ and rendered image $G(w, c)$, we utilize VGG16 image encoder $V(\cdot)$ [52] to extract the image features and construct the Reconstruction Loss as following:

$$\mathcal{L}_R(G(w,c), x) = ||V(G(w,c)) - V(x)||_2^2 \tag{2}$$

For human faces, we adopt the same Identity Loss as [46]:

$$\mathcal{L}_{ID}(G(w,c), x) = 1 - \langle R(x), R(G(w,c)) \rangle \tag{3}$$

where $R$ is the pretrained ArcFace [14] network. VGG16 image encoder and ArcFace $\mathcal{L}_R$ and $\mathcal{L}_{ID}$ are only used to measure the difference between the input image and the rendered image from input camera's view to avoid the misalignment due to mismatched viewing directions. See our ablation study for an insightful analysis of the interaction between the Reconstruction and Identity losses.

### 3.4   Prompt Editing with Diffusion Model

Although directly optimizing a NeRF by distilling a 2D diffusion model is possible [44, 32], the quality of the resulting model is often worse than the used 2D model, especially in the domain of human subject (refer to Appendix C.2). In addition, compared with 3D diffusion models, 3D GANs currently are capable of producing high-fidelity 3D NeRFs with a smooth latent space. Therefore, we perform Score Distillation in the latent space of trained 3D GAN model.

We modify the Score Distillation Sampling (SDS) from DreamFusion [44] as our diffusion loss. The loss connects the $x'$ rendered from a sampled camera pose $c_s$ with the denoising prediction conditioned on the editing prompt $y$. The denoising process ca n be written as:

$$\mathcal{L}_D(x' = G(w, c_s), y) = \mathbb{E}_{\varepsilon(x'),y,t,\epsilon} \left[ \| \hat{\epsilon}_\phi(\mathbf{z_t}; y, t) - \epsilon \|_2^2 \right] \tag{4}$$

where $\varepsilon(\cdot)$ is the image encoder that encodes our rendered image $x' = G(w, c_s)$ to the latent space of the diffusion model, denoted as $z$. Here $z_t$ is the noisy version of $z$ at time-step $t$, $\hat{\epsilon}_\phi$ is the frozen denoising network of the trained diffusion model, where we sample $t \sim \mathcal{U}(0.02, 0.98)$ to avoid very high and low noise levels; $\epsilon \sim \mathcal{N}(0, I)$, and its effect on input latent varies with time-step $t$, which is same as done in Dreamfusion [44].

Notably, the camera pose $c_s$ will be resampled randomly in each optimization iteration to ensure 3D view consistency. Unlike DreanFusion, our text prompt $y$ is view independent, since we have the identity and reconstruction losses to constrain the optimization process, and that the EG3D generator has underlying 3D information in contrast to randomly initialized NeRF. Therefore, our diffusion loss back-propagates to the latent scalars, where the gradient of $\mathcal{L}_D$ is given by

$$\nabla_w \mathcal{L}_D(x' = G(w, c_s), y) \triangleq \mathbb{E}_{\varepsilon,t,\varepsilon} \left[ w(t)(\hat{\varepsilon}_t(\mathbf{z_t}; y, t) - \epsilon)\frac{\partial x'}{\partial w} \right] \tag{5}$$

As in Dreamfusion [44], $w(t)$ absorbs the coefficient of the forward process, and we ignore the term $\frac{\partial \hat{\varepsilon}_t(\mathbf{z_t}; y, t)}{\partial \mathbf{z_t}}$ as the diffusion model is frozen.

### 3.5   Explicit Illumination Control

As the trained 3D GAN mapped a continuous latent code to a 3D-consistent NeRF, direct manipulation of the latent ensures view-consistent illumination editing.We utilize [72] to construct our Illumination Loss, where the hourglass network can be denoted as:

$$\mathbf{L}_s^*, \mathbf{I}_t^* = \mathbf{Hn}(\mathbf{L}_t, \mathbf{I}_s) \tag{6}$$

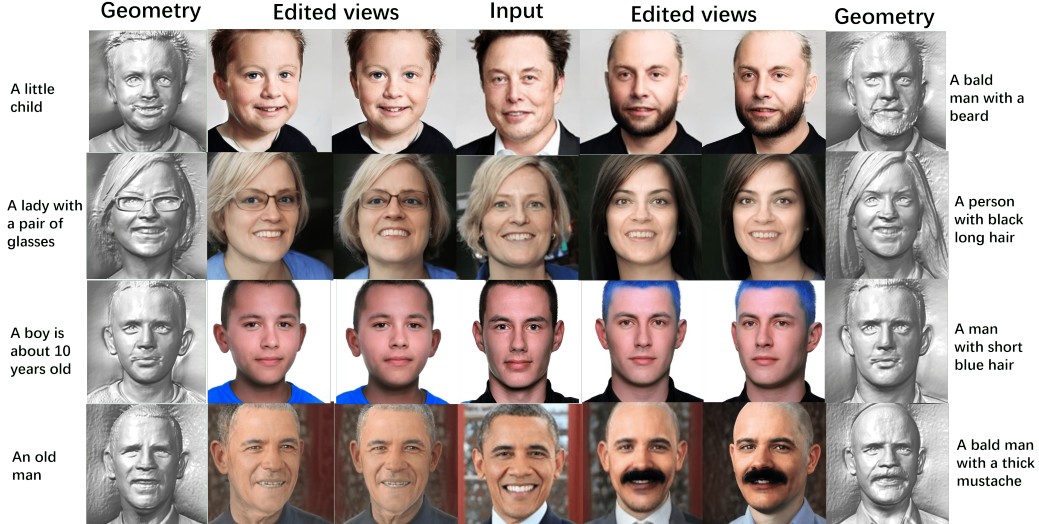

| Geometry | Edited views | Input | Edited views | Geometry |

A little child

A lady with a pair of glasses

A boy is about 10 years old

An old man

A bald man with a beard

A person with black long hair

A man with short blue hair

A bald man with a thick mustache

Figure 3: **More FaceDNeRF results.** The middle column shows the input images, while the left and right halves of the images in the other two columns show the results of text prompts editing on the left and right, respectively. The first and last columns show the corresponding geometries.

where $\mathbf{L}_t$ and $\mathbf{L}_s^*$ are respectively the target SH lighting and the estimated SH lighting of the input image $\mathbf{I}_s$, and $\mathbf{I}_t^*$ is the relit image under the target SH lighting. Here, we only use $\mathbf{L}_s^*$ to compute the $L_1$ loss with $\mathbf{L}_t$, and $\mathbf{I}_t^*$ will be ignored, i.e., $\mathbf{L}_s^* = \mathbf{Hn}'(\mathbf{I}_s)$. Thus, our Illumination Loss is:

$$\mathcal{L}_{IL}(G(w, c_s), l_{c_s}) = \| \mathbf{Hn}'(G(w, c_s)) - l_{c_s} \|_1 \tag{7}$$

During the optimization process of $w^*$, differentiability is crucial. Given $G(w, c_s)$ is differentiable with respect to $w$, the differentiability of $\mathbf{Hn}'(G(w, c_s))$ with respect to $G(w, c_s)$ guarantees the back-propagation onto the $w$. Therefore, we replace the PyTorch in-place operations and other numpy operations in the $\mathbf{Hn}'$ model with equivalent differentiable PyTorch tensor operations. As verified by experiments, our modified $\mathbf{Hn}'$ achieves the same performance as the original model while being multi-view consistent as we render the output.

### 3.6 Pivotal Tuning Inversion

The Pivotal Tuning Inversion (PTI) technique [47] is widely employed in many GAN inversion processes to overcome the trade-off between distortion and editability [58]. After solving the Eq. (1), the obtained latent code $w_t$ is utilized as a pivot to fine-tune the generator $G$ using the similar loss function:

$$\arg\min_{G} \ \lambda_{ID}\mathcal{L}_{ID}(G(w_t, c), x) + \lambda_R\mathcal{L}_R(G(w_t, c), x) + \lambda_D\mathcal{L}_D(G(w_t, c_s), y) + \lambda_{IL}\mathcal{L}_{IL}(G(w_t, c_s), l_{c_s}). \tag{8}$$

## 4 Experiments

This section presents our editing results and compare them with representative methods, emphasizing FaceDNeRF's disentanglement capability which is conducive to editing control, its flexibility in utilizing a text-guided diffusion model, as well as the multi-view consistency in illumination control. Furthermore, our method can be migrated to other data domains with trained generative models.

### 4.1 3D Editing from Single Image

Although diffusion models can generate diverse and realistic images in 2D, the lack of large and high-quality 3D datasets limits the performance of current diffusion models. On the other hand, the adversarial learning mechanism together with inductive bias of 3D rendering enables GAN to produce high-quality 3D results. From GAN's smooth latent space, our method finds the suitable latent code whose semantic information matches the editing demand. Fig. 1 and Fig. 3 demonstrate the detailed editing results of FaceDNeRF. Please refer to Sec. B of the appendix for more results.

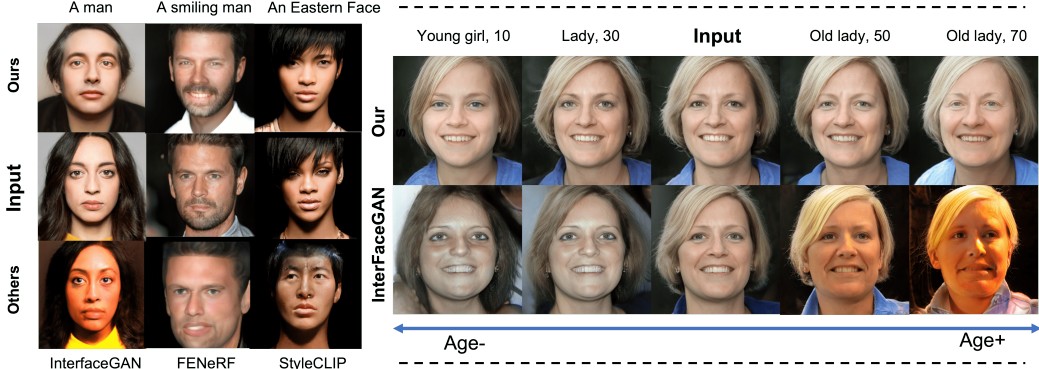

Figure 4: **Editing Comparison with representative methods**. We compare with classifier-based InterfaceGAN [51], semantic edited FENeRF [57] and language guided StyleClip [42] with EG3D. Images on the left side are editing results of gender, smile, and "An Eastern face" respectively. The right images are age editing comparison with [51], and our input text prompt are "A *XX* is about *YY* years old", where *XX* and *YY* are depicted above. The comparison illustrates the flexibility and high-fidelity editing capability of FaceDNeRF.

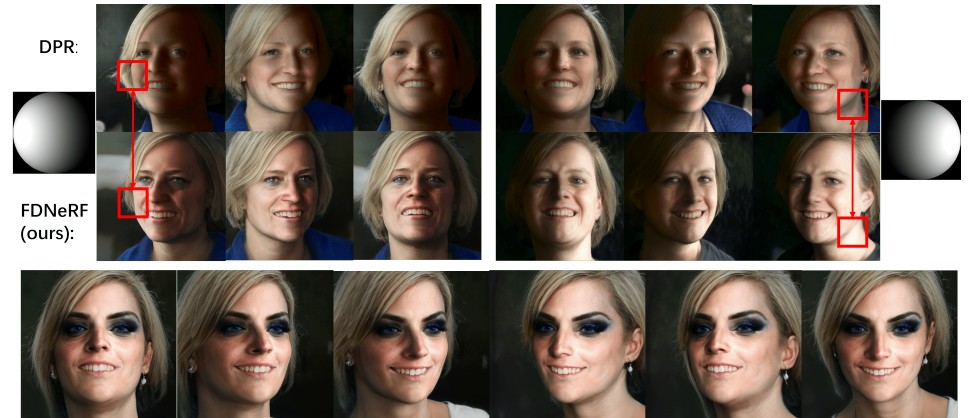

Figure 5: **Illumination Comparison**. The images in the first row are relit images generated by DPR [72]. The images in the second row are relit images generated by our method. Upon comparing the four red boxes, our method is observed to produce fewer artifacts and more realistic results. Furthermore, the images in the third row demonstrate that our method can not only edit face NeRF using text prompts but also allows explicit control of lighting.

## 4.2 Comparison

**Comparison with representative editing methods**     For effective and easy editing, the underlying latent space should be smooth and semantically meaningful. Compared with the latent space of 2D GANs, disentangling in 3D is much harder. InterfaceGAN [51] seeks a hyper-plane in the latent space with pre-trained classifiers, which makes editing feasible by interpolating latent codes in the direction orthogonal to the hyper-plane. However, the assumption that a good hyper-plane exists which is well-behaved for linear interpolation is only valid when the latent space is highly disentangled. As shown in Fig. 4, the underlying latent space in 3D is not well disentangled. Thus the interpolation (induced by editing) can interfere and affect other irrelevant attributes not to be edited. We also compare with the semantic-based method FENeRF [57], whose editing is performed by manually editing the semantic map. However, this representation limits editable attributes, especially on semantically complex attributes such as age and gender. In addition, we compare with language-guided StyleClip [42] using the same latent space. Our method achieves better editing quality conditioned on the same text prompt, indicating the superiority of utilizing the diffusion model as guidance. Please refer to

| A bald man | A woman with blond hair | A man with mustache wearing a pair of black glasses | A sad kid | A black car | A white cat with blue eyes |

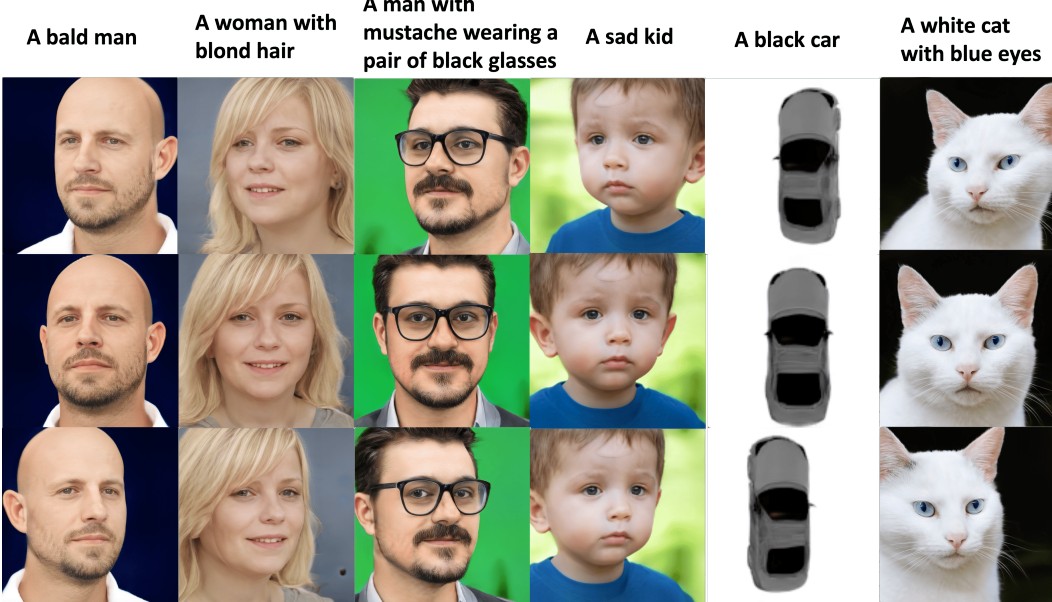

Figure 6: **Text-driven 3D Generation**. These images are generated and conditioned solely on text prompts. We utilize the trained EG3D generators [8] from different data domains.

Appendix Sec. C for more qualitative and quantitative comparisons, and insights into why SDS guidance outperforms CLIP guidance.

**Illumination Comparison**     There have been attempts in 2D to control illumination, either implicitly or explicitly. To produce results compatible in 3D, a straightforward approach is rendering multi-view images from a given NeRF, followed by using the projected lighting direction for each view to render the edited NeRF. However, as shown in Fig. 5, without the constraint of 3D rendering, inconsistent illumination is easily observed, indicating imperfect alignment using 2D methods. With explicit 3D control, FaceDNeRF directly manipulates latent code in the disentangled space, capable of rendering consistent and realistic lighting results.

**Migration to Other Data Domain**     FaceDNeRF is not limited to human faces: the semantically diverse and rich text-guidance diffusion model can be used to directly edit other data domains. As shown in Fig. 1, we perform editing on GANs trained on Cats and Cars, noting the reconstruction loss is also universal to all data domains.

### 4.3    Text-conditioned 3D Generation

In addition to editing, FaceDNeRF can generate high-quality 3D models given a text prompt. Similar to Dreamfusion [44], we guide the generation process under the iterative supervision of a trained diffusion model. However, their unconstrained optimization with random NeRF initialization lacks the ability to generate realistic results. Fig. 6 shows some generated high-quality examples by our methods, including examples from other data domains.

**Latent Regularization**     Although the latent space is smooth, generation from rare-sampled latent codes may produce unrealistic results. Latent samples around mean latent code tends to give better outputs (truncation trick in StyelGAN2 [27]). In text-conditioned 3D generation, the optimized latent code is more likely to deviate from the latent mean since there is no reconstruction or identity restriction. Thus we add a latent regularization to encourage results around the mean, formulated as:

$$\mathcal{L}_{regu}(w, \bar{w}) = \lambda_{regu}\|w - \bar{w}\|_2^2 \tag{9}$$

where $\bar{w}$ is the sample mean of latent space. This regularization is equivalent to constraining the feasible space around the mean latent code (Lagrange Multiplier).

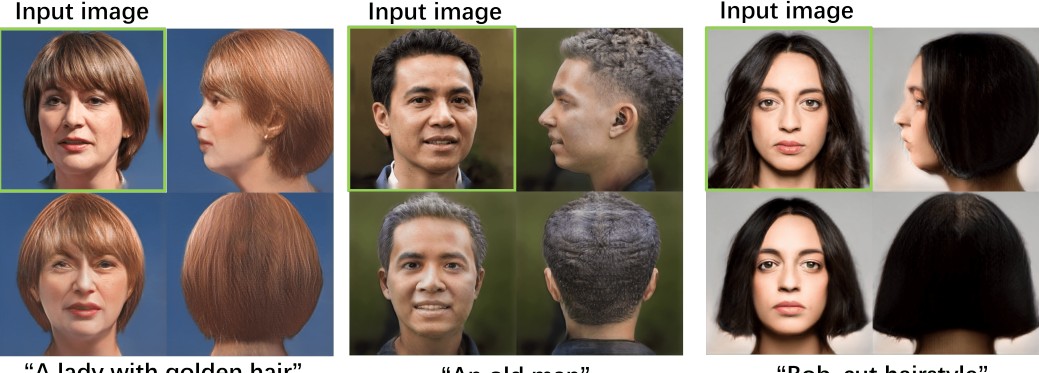

Input image    Input image    Input image

"A lady with golden hair"  "An old man"  "Bob-cut hairstyle"

Figure 7: Results on replacing the EG3D backbone with PanoHead backbone. Top-left sub-image in the respective three sub-figures is the input image, with the input text prompt below each subfigure.

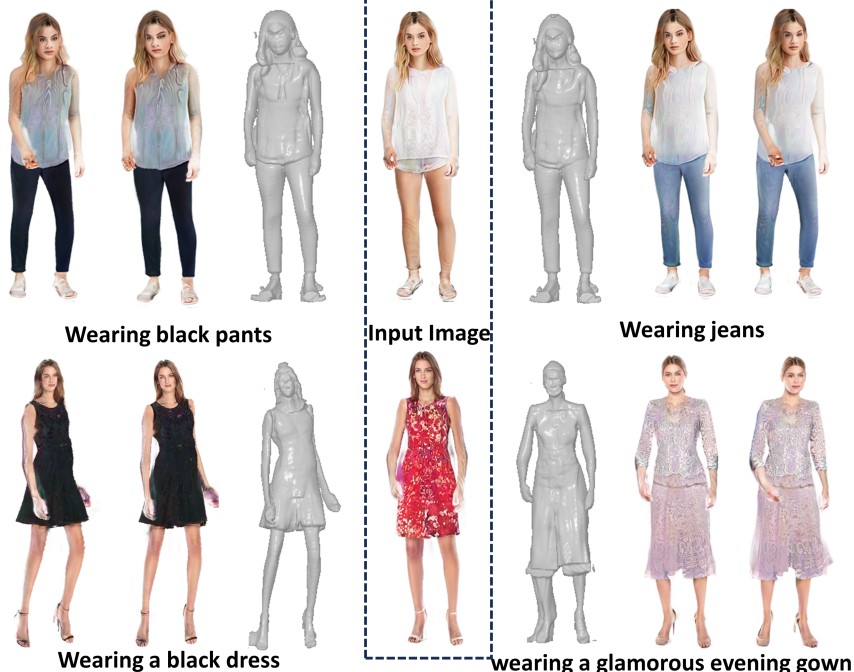

Wearing black pants   Input Image   Wearing jeans

Wearing a black dress   wearing a glamorous evening gown

Figure 8: Semantics-driven editing on human body by replacing EG3D with EVA3D, a state-of-the-art human body NeRF generator. Each row shows 2 examples with the middle image as input.

## 4.4 Generalizability Across Different Backbones & Data Domain

In addition to utilizing EG3D [8] as our backbone, the optimization process of FaceDNeRF can be effectively applied to alternative backbones.

**Face Domain Generalization** In this instance, we exhibit the implementation of PanoHead [3] as the chosen backbone. Given that PanoHead operates as a generative model for human facial representations, akin to the EG3D, it allows for the smooth integration of FaceDNeRF. The outcomes are illustrated in Fig. 7 .

**Cross-Domain Generalization** Here we demonstrate the extension of our method to the human body domain, which is challenging due to the articulation of human bodies. We employ EVA3D [21], a cutting-edge human body NeRF generator for optimization. Notable artifacts are still present in state-of-the-art methods like EVA3D. As the latent variable deviates from the mean value, the generated result becomes blurry, and the face becomes unrecognizable. Fig. 14 in the appendix

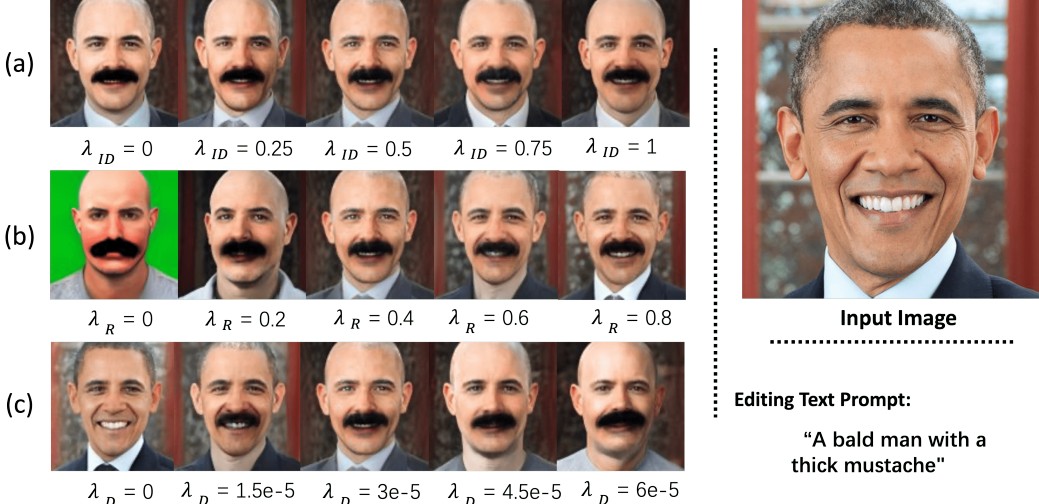

Figure 9: **Ablation studies on $\lambda_{ID}$, $\lambda_R$, and $\lambda_D$.** Input text prompt is "A bald man with a thick mustache". The central images in each row are identical with the setting ($\lambda_{ID}$, $\lambda_R$, $\lambda_D$) = (0.5, 0.4, $3 \times 10^{-5}$). Each row shows the impact of changing one of the weights associated with a particular loss term.

displays generation outcomes by random sampling in the latent space near its mean value, revealing a noticeable lack of 3D consistency and erroneous geometric estimation even near the mean value. Fig. 8 illustrates the editing results. Due to EVA3D's limited generation capacity, we conduct editing on images it generated from randomly sampled latent variables close to the mean value of the latent space. Specifically, we randomly sample two values in the latent space, averaging them to obtain a latent variable value. During each iteration, we calculate the feature loss using the input image and a rendering of the current human body NeRF from the same viewpoint. The diffusion loss and optimization pipeline remain identical to our EG3D version. As depicted in Fig. 8, despite EVA3D's suboptimal generation quality, our editing remains flexible and reasonable. High-quality NeRF generators, inversion methods, and advanced feature or identity preservation techniques are potential future works that could enhance the generalization of our pipeline to other domains.

## 4.5 Ablation Study

To assess the impact of the weights of various losses on the generated NeRF, we perform ablation studies on the weights of Identity Loss $\lambda_{ID}$, Reconstruction Loss $\lambda_R$, and Diffusion Loss $\lambda_D$. The frontal-view renderings of the NeRF results from these studies can be seen in Fig. 9. For a comprehensive analysis, please refer to Sec. D.1 in the appendix. Moreover, we investigated the impact of the editing prompt on editing outcomes. For results and a detailed discussion, please see Sec. D.2 in the appendix.

## 5 Conclusion and Discussion

We propose FaceDNeRF, a new generative method to reconstruct high-quality Face NeRFs from a single image, with semantic prompt editing and relighting capitalizing on recent stable diffusion contributions. We believe our optimization pipeline starts a new approach in semantics-driven editing and relighting given any NeRF GAN, and thus has a good impact and can spawn worthwhile future work in the years to come. Extensive experiments validate our significant improvement over state-of-the-art 3D face reconstruction and editing methods. The proposed FaceDNeRF is readily applicable to many real-world applications such as 3D face manipulation, which, however, might be used unethically. Also, the upper bound of our performance is limited by the chosen GANs or diffusion models. We leverage EG3D [8], which is trained on real-world datasets, thus our generation results are realistic but confined to real-world faces or objects, FaceDNeRF can go beyond faces and can be extended to a generic NeRF generation and editing template, where different 3D generators can replace EG3D [8] to produce NeRFs in various domains with ease.

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

# Appendix

## A    Implementation Details

Our reconstruction loss and identity loss are applied to the ground truth image camera poses. Due to our limited GPU memory, we only render one side view to calculate the diffusion loss and illumination loss at each iteration. The camera rotation angles $\theta$ and $\phi$ are randomly sampled from $\left[\frac{\pi}{2} - \frac{\pi}{12}, \frac{\pi}{2} + \frac{\pi}{12}\right]$ and $\left[\frac{\pi}{2} - \frac{\pi}{12}, \frac{\pi}{2} + \frac{\pi}{12}\right]$, where $\theta$ and $\phi$ are the angles of spherical coordinate. We set the optimization iterations for our editing to 500, which takes approximately 10 minutes on a 3090 GPU. We set the weighting $\mathcal{L}_D, \mathcal{L}_{ID}, L_R$ to be 0.2, 0.2 and $2 \times 10^{-5}$ for most editing cases, which can be finetuned for each editing.

**Dataset and Generative Bias**    We utilize trained checkpoints of EG3D on FFHQ [25], AFHQv2 [11] and ShapeNet [9] for the data domain of face, cat and car respectively. For some editing and generation, we notice the existence of biases in generated results caused by the bias of the training dataset, especially for race, gender, etc.

## B    More Results

As a supplement to Fig. 1 and Fig. 3 in our main paper, Fig. 10 shows more results of our FaceDNeRF. All three figures illustrate that given a single face image, our FaceDNeRF can perform semantics-driven NeRF editing on various features, such as expressions, emotions, glasses, hairstyles, races, genders, ages, makeup, beard, mustache, and goatee. Notably, both individual and joint editing of these features can be achieved in high fidelity.

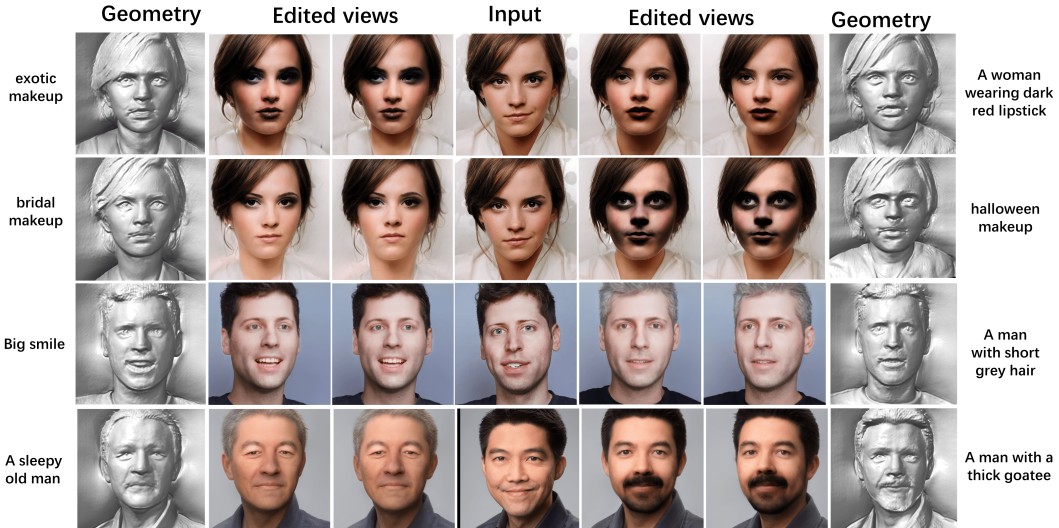

Figure 10: **More FaceDNeRF results.** The middle column shows the input images, while the left and right halves of the images in the other two columns show the results of text prompts editing on the left and right, respectively. Additionally, the first and last columns showcase the corresponding geometries.

**Text-conditioned Generation Comparison**    Our attempts to compare with Dreamfusion[44] (Implemented on a StableDiffusion) failed since it cannot generate faithful models for human heads. This problem might be caused by various reasons. First, the ambiguity of text prompts (human identity, rendering directions) might result in an inconsistent denoising behavior at each iteration. Also, the randomized and unconstrained NeRF optimization process might collapse. On the contrary, our utilization of latent code and trained 3D generative models ensures successful and high-quality generation.

# C   More Comparisons

## C.1   InterfaceGAN and StyleCLIP

Here we provide supplementary comparisons (both qualitative and quantitative) with InterfaceGAN and StyleCLIP. In Fig. 11, since StyleClip proposed two methods Latent Optimization and Latent Mapper, we compared them separately. We adopt the same text example as Fig.4 in the StyleClip paper. For clarity, we only present the results of our method and the Latent Optimization method. Please refer to the original paper for the results of the Latent Mapper.

The qualitative results demonstrate that our method outperforms the Latent Optimization method in terms of consistency with the text prompt and identity preservation, which are also corroborated in the quantitative results in Tab. 1. It is important to note that the Clip loss alone cannot fully capture the visual perception of distances between images and the semantic meanings of text prompts. In some cases (e.g., Fig.1, 1st-row 3rd-column, 3rd-row 3rd-column, and 4th-row 5th-column), the results of the Latent Optimization method exhibit degradation while the Latent Optimization method achieves the lowest Clip loss. On the other hand, our method consistently produces reasonable results across all cases. Even when compared to the latent mapper method, which requires 10–12 hours of training for a specific text input, our method achieves a comparable level of visual quality. Our method only needs around 10mins inference time and no training is needed for a specific text prompt. (Times are measured on a single NVIDIA GTX 1080ti GPU.)

As shown in Fig. 12, we compare with InterfaceGAN on more attributes. Note that as the latent direction found by the classifier could be entangled, the editing results of InterfaceGAN often exhibit undesired changes in other attributes which leads to identity changes. Tab. 2 quantitatively verifies that our method preserves Identity better while achieving the target editing.

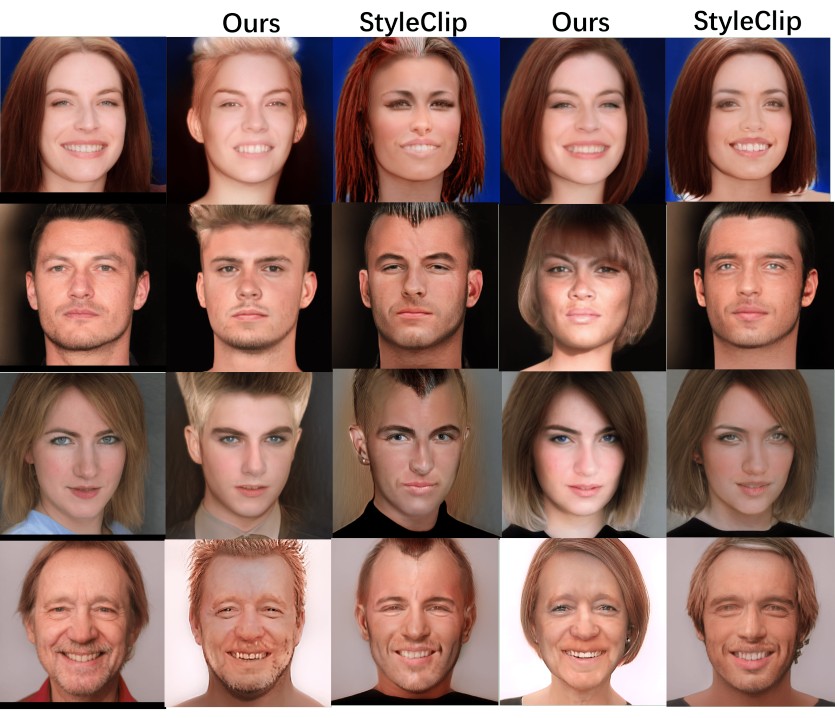

Figure 11: Qualitative comparison of FaceDNeRF with StyleClip (latent Optimization). To mitigate the influence of various weight choices on the final outcomes, we experimented with 40 different combinations encompassing a wide range of weights. We present the results with $IDLoss \leq 0.05$ and the highest visual consistency with the given text prompt.

Table 1: (a) The Latent Optimization method in StyleClip paper, (b) The Latent Mapper method in StyleClip paper. Ours, (a), and (b) are evaluated on the examples shown in Fig. 11. The values of ID loss are multiplied by 100. The values of Clip loss are multiplied by 10.

| | Ours | (a) | (b) |
|---|---|---|---|
| ID Loss | 2.26±1.26 | 2.40±1.52 | 1.68±0.60 |
| Clip Loss | 7.49±0.24 | 6.53±0.17 | 7.16±0.19 |

Table 2: ID loss on InterfaceGAN and Our FaceDNeRF. We note that the latent direction for InterfaceGAN be may entangled, which causes more alternation in Identity.

| | Ours | InterfaceGAN |
|---|---|---|
| Young | **0.0102** | 0.0159 |
| Smile | **0.0028** | 0.0105 |
| Mustache | **0.0014** | 0.0148 |

## C.2 SDS in Latent Space

To show the usefulness of using SDS as guidance in GAN latent space, we provide comparison with directly using the original SDS on unconstrained NeRF space. As shown in Fig. 13, the original SDS can only generate a blurry human head, where the process can collapse with less careful prompts. This is because of the over-saturation and low-diversity problems of SDS. There are methods to directly apply diffusion in 3D human heads (Rodin), yet they rely on 3D ground truth data (synthesized), which limits the quality and diversity of 3D head diffusion models, while GAN requires no specific 3D data and achieves high-quality 3D results. Therefore, with trained GAN as a constraint, our SDS can effectively utilize the latent space.

## C.3 Using CLIP as Guidance

In this subsection, we will discuss why SDS guidance outperforms CLIP guidance significantly. During our comparison with StyleCLIP, we empirically found that the CLIP loss tends to cause degraded images (i.e., images with low image quality and obvious artifacts, not similar to the original image, not coherent to the text prompts). We suspect several reasons for this phenomenon.

During the optimization process using CLIP guidance, the optimization is based only on the current local gradient estimated by the cosine similarity between text embedding and image embedding, we empirically observe that it may overshoot to degraded images. In the SDS setting, the objective is to find the most probabilistic mode in the image domain conditioned on the text, which is referred to as mode seeking [73]. Also, the t sampled will determine the weight of diffusion guidance, which affects the magnitude of the gradient. This offers a more robust updating process, reducing the possibility of being trapped in a local minimum.

In addition, diffusion models focus on high-quality generation, while CLIP focuses on learning cross-modal representations for understanding the relationship between images and text. However, as discussed in [6], CLIP and many other multi-modal understanding models suffer from 'bag-of-objects' kind of representations, possibly due to the contrastive pre-training settings. They perform well in understanding nouns, but badly in visual language concepts beyond object nouns, like attributes and relations. In our context, the input text prompt involves many non-object concepts, like emotions, colors, and other attributes of faces and decorations (e.g., "blue" hair, a "bald" man, an "Eastern" face, a "manly" woman, an "angry" lady, a woman "wearing" a pair of glasses, etc.). When using CLIP guidance, the rendered image is encoded to calculate cosine similarity with the text embedding, during which spatial and attribute information in the rendered image are lost, with a focus on nouns. Thus, CLIP guidance easily "losses control" over the rendered images and generates degraded images, due to difficulty in understanding non-object concepts. In contrast, noise is injected during the training of diffusion models, which allows the denoiser to incorporate more spatial information during the iteration process. More concepts beyond object nouns are also considered by diffusion models. Therefore, we believe that SDS guidance provides more strict "control" and thus higher quality guidance.

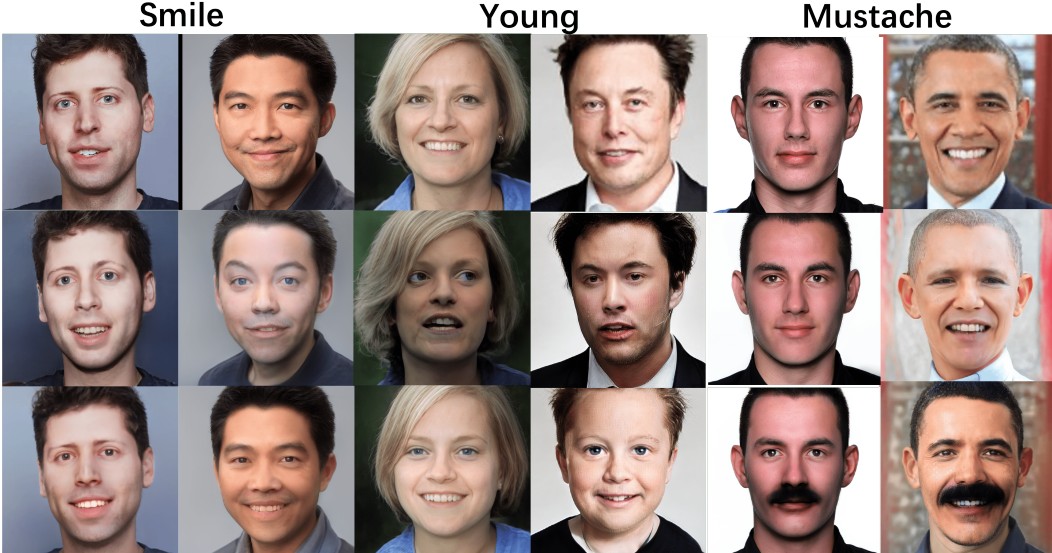

Figure 12: Comparison with InterfaceGAN. From the top, rows are respectively input, InterfaceGAN results and our results. We can observe in our results better identity preservation and more faithful edited result to the desired attributes.

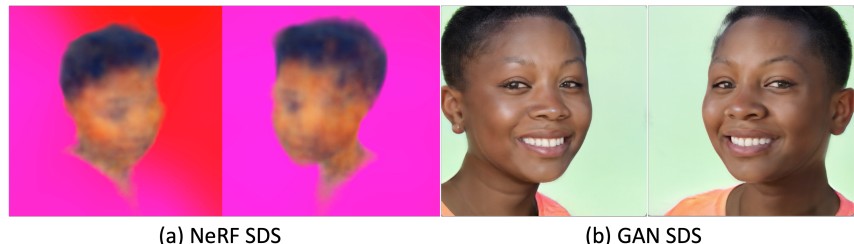

(a) NeRF SDS                                    (b) GAN SDS

Figure 13: Comparison of SDS on GAN Latent Space with SDS on unconstrained NeRF space. (a) The original SDS operates on unconstrained NeRF space but can only generate blurry human heads which can even fail in some cases. (b) High-quality results generated from GAN latent space guided by SDS.

# D   Ablation Study

## D.1   Loss Weights

To investigate the different losses on the generated NeRF, we conduct ablation studies on:

(a) Weight of Identity Loss $\lambda_{ID}$;

(b) Weight of Reconstruction Loss $\lambda_R$;

(c) Weight of Diffusion Loss $\lambda_D$.

Fig. 9 shows the frontal-view rendering of the resulting NeRF in our ablation studies. The input text prompt is "A bald man with a thick mustache". The central images in each row are identical, adopting $(\lambda_{ID}, \lambda_R, \lambda_D) = (0.5, 0.4, 3\times10^{-5})$. In each row, only one weight is changed to show the effect of the corresponding loss term.

Observing the rendering results in row (a) and row (b) in Fig. 9, the results with either larger $\lambda_{ID}$ or larger $\lambda_R$ look more similar to the input person, indicating that both Identity and Reconstruction Losses help preserve the person's identity, while the two losses operate in different ways.

Identity Loss is calculated using pre-trained ArcFace [14] network as described in Sec. 3.3 in our main paper. ArcFace [14] trains DCNNs for face recognition which maps the face image into a feature with small intra-class but large inter-class distance, which means a person with different expressions, hairstyles, mustache, or subtle wrinkles, should have a similar identity score, thus low

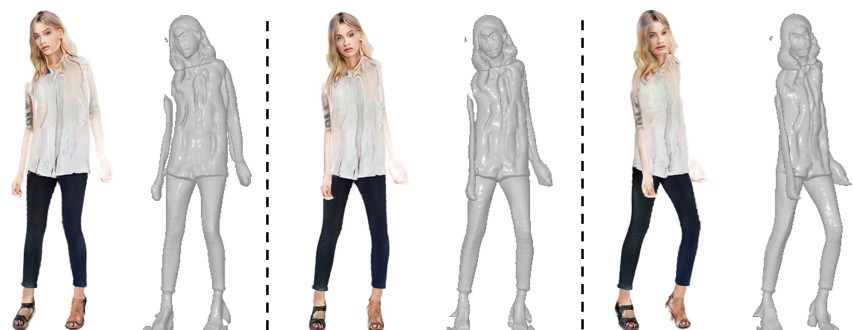

Figure 14: Generation Results using EVA3D by randomly sampling around the latent space's mean value. Despite the latent variable for generation being close to the mean value of the latent space, there is still a noticeable lack of 3D consistency and visible geometric estimation artifacts.

Identity Loss in our case. Therefore, Identity Loss would not encode details such as hairstyles and subtle wrinkles, but help to preserve large-scale identity-specific information, such as the shape of the head, nose, mouth, eyes, eyebrows, and any apparent wrinkles.

Instead, Reconstruction Loss calculated by Eq. (2) in our main paper takes all pixels into consideration, thus trying to preserve all kinds of features indiscriminately. A large value of $\lambda_R$ results in a face NeRF similar to the input image. However, this dilutes the guidance effect from the text prompt's attempt to edit the NeRF by removing hair and adding a mustache in Fig. 9. From the right two images of row (b) in Fig. 9, we observe that the identity still has apparent hair, since the large weight of Reconstruction Loss forces the face NeRF to be as similar as possible to the input image.

This is a trade-off between fidelity to details in the input image versus fidelity to the text prompt. As a result, our joint utilization of Identity Loss and Reconstruction Loss can significantly alleviate the problem, where Identity Loss can preserve the identity information in a less contradictory way in the presence of text guidance.

Results in row (c) of Fig. 9 shows the effect of $\lambda_D$. A larger value of $\lambda_D$ enhances guidance from the diffusion model, while a smaller value of $\lambda_R$ produces higher-fidelity results to the input image. Users could choose suitable value of $\lambda_{ID}$, $\lambda_R$, and $\lambda_D$ based on the specific task.

### D.2   Editing Prompts

We investigate the influence of the input text prompt in this section. In Fig. 10, and Figure 1 and 3 in our main paper, we mainly show NeRF editing results when the input text prompt is a short sentence or a full description of the expected NeRF, such as "A woman wearing dark red lipstick". While this helps a valid generation that avoids ambiguity in semantics, our FaceDNeRF can also take in a text prompt that only expresses the difference between the input face and the output face, such as "Dark red lipstick", if the input face has no lipstick at all. Here, Fig. 15 shows the ablation results. The right two columns show NeRF editing results when input text prompts are full descriptions of the expected editing results, while the left two columns are results with text prompts only describing what should be different.

In most cases, our FaceDNeRF can generate similar editing results given either type of text prompt with subtle or even no tuning of weights of losses $\lambda_{ID}$, $\lambda_R$, and $\lambda_D$, as shown in Fig. 15. However, sometimes ambiguity in semantics interferes editing when the input text prompt is not a full description of the expected output. As shown in Fig. 16, even though our FaceDNeRF can generate an old Elon Musk given "An old man" as the input text prompt, it fails when the input becomes just "Old". We believe this is because "Old" has various meanings depending on its context. Therefore, a single "Old" without any context results in useless guidance from our diffusion model, thus producing poor editing results. "Elderly" and "senior" are two synonyms of "Old" in this context. As shown in Fig. 16. "senior" also fails due to its semantic ambiguity with no context, while "elderly" succeeds because of its specific meaning.

Thus, users of our FaceDNeRF are advised to give a better and complete text prompt to avoid semantics ambiguity.

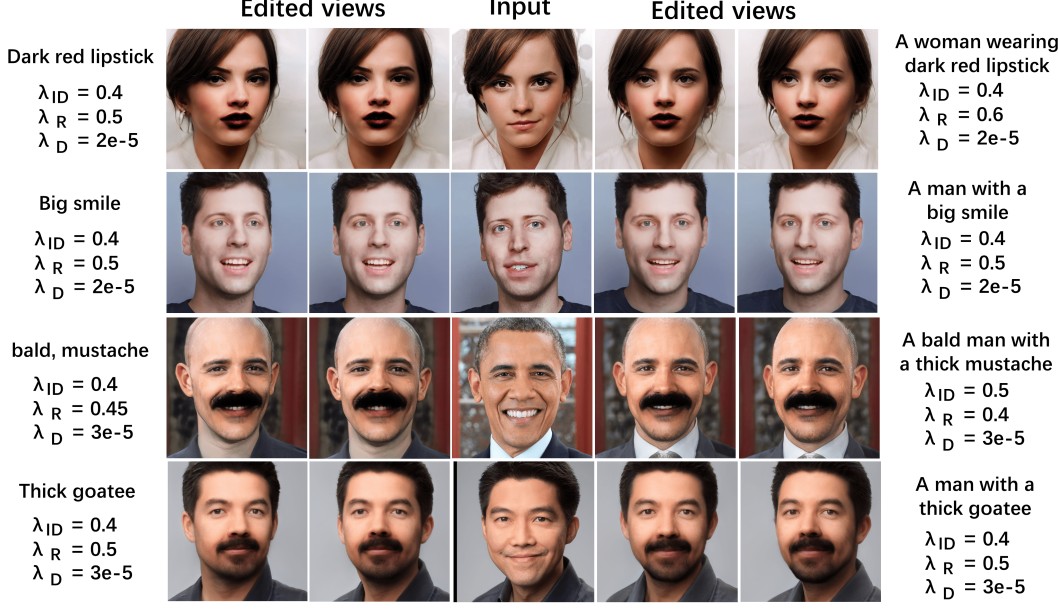

Figure 15: **Ablation study on the text prompt.** On the right are results generated with full text prompts. The left ones are produced using simlfied prompts. Given text prompt describing the wanted change, FaceDNeRF generates results with desired editing.

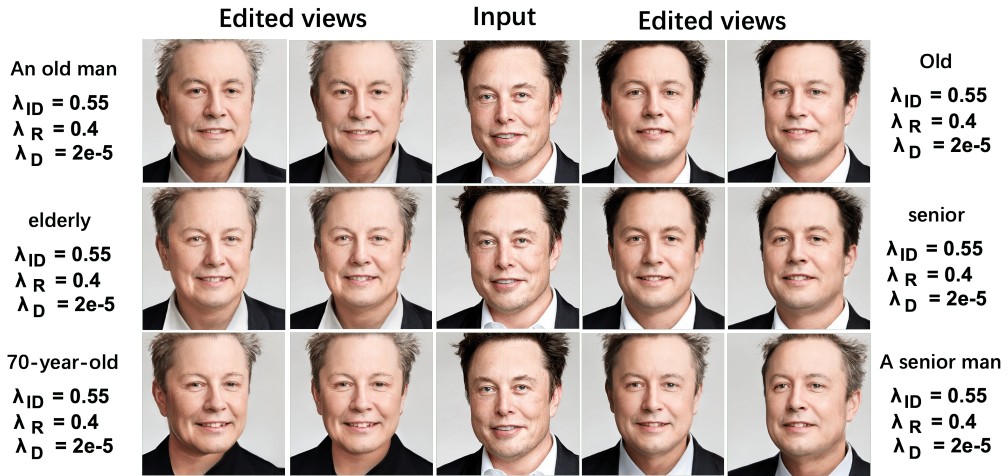

Figure 16: **Ablation study on synonyms.** Here are results generated with various Synonyms, which could affect the results. The contest is important for synonyms to eliminate ambiguity, as the case of "senior" and "A senior man".

