# OpenReview forum: "FaceDNeRF: Semantics-Driven Face Reconstruction, Prompt Editing and Relighting with Diffusion Models"
_NeurIPS.cc/2023/Conference — NeurIPS 2023 poster_

### Official Review · Reviewer_4TUt · 2023-06-21

**Soundness:** 2 fair
**Presentation:** 2 fair
**Contribution:** 1 poor
**Rating:** 4
**Confidence:** 5

**Summary:**

The paper introduces FDNeRF, a generative method that reconstructs Face NeRFs with semantic editing and relighting capabilities. FDNeRF utilizes 3D GAN inversion and a trained 2D latent-diffusion model to enable zero-shot learning, eliminating the need for explicit 3D data. With designed loss functions and multi-modal pre-training, FDNeRF offers extensive control over editing processes using single-view images, text prompts, and target lighting. Experimental results demonstrate the realistic outcomes and editing flexibility of FDNeRF.

**Strengths:**

+ It includes a illumination loss using SH and differentiable renderer to preserve the lighting condition.
+ The illumination loss allows explicit control over the lighting in a view-consistent manner.

**Weaknesses:**

- I found it challenging to gain valuable insights from this paper as it primarily combines conventional losses, including the identity loss, SDS loss, reconstruction loss, and illustration loss. From a research perspective, the technique presented in this paper offers limited contributions.

- While the paper showcases facial images from various viewpoints, it fails to evaluate its performance on larger poses. The current results are restricted to angles between -30 to 30 degrees, making it difficult to fully comprehend the effectiveness of this approach for 3D reconstruction and editing tasks.

- The paper lacks a comprehensive evaluation of the inversion process. It is crucial to compare the proposed inversion method with state-of-the-art EG3D inversion methods to establish its performance. Additionally, based on the information presented, the inversion method implemented in this paper appears to be inadequate and could benefit from further refinement.


**Questions:**

- Figure 3, in particular, leaves me perplexed. The prompt-guided editing appears to completely alter the original facial attributes, focusing more on generating new features rather than preserving the original identity.

- I found it challenging to assess the effectiveness of the diffusion loss utilized in this model. It would be beneficial to understand the advantages of the SDS loss in comparison to CLIP guidance. A clear demonstration of the superiority of the SDS loss over other techniques would provide valuable insights.

-

**Limitations:**

The author should add a figure to illustrate the limitations.

---

> ### Author Rebuttal · Authors · 2023-08-10
>
> We thank you for your thoughtful reviews and suggestions. We are encouraged that you appreciate the two strengths of our work.
>
> As explained in responses to common concerns, we extended our method to 360 human heads based on PanoHead. We also utilize PTI for the inversion process. The results of Fig. 2 have already incorporated the PTI fine-tuning.
>
> We have a known trade-off between identity preservation and degrees of editing in many editing frameworks. As quantitatively and qualitatively evaluated in the global response, our method achieves better identity preservation while flexible for attribute-guided editing. The trend to generate new images may be attributed to several reasons: First, the prompts in our editing are simple (which makes editing easier) and carries no identity information. Therefore, the diffusion model will inevitably generate different results, which makes the usage of ID loss and VGG loss crucial. Second, as in the naive SDS case, the SDS loss tends to be over-saturating. The ID-preservation losses ameliorate this problem.
>
> For the CLIP loss, during our comparison with StyleCLIP, we empirically found that the CLIP loss tends to produce degraded images (i.e., images with low image quality and obvious artifacts,  not similar to the original image, not coherent to to the text prompts such as examples of 1st-row 3rd-column and 4th-row 5th-column in Fig.1). We suspect several reasons for this phenomenon. First, the denoising process of SDS adds noise at each iteration, which reduces the possibility of being trapped in the local minimum. Second, the CLIP loss relies on cosine similarity, which lacks knowledge of how the current updated step will affect the image. Thus, these possible causes may deviate from the original image drastically. On the contrary, the reverse process (denoising) of SDS tries to minimize the score of the current image conditioned on the text prompt at all iterations, which provides more consistent guidance.
>
> We argue that our choice of framework is not a simple combination of existing losses. Our motivation to utilize SDS loss in the GAN latent space is justified, together with ID loss and VGG loss that preserves identity and illumination loss that controls the lighting condition explicitly, our methods achieve high-fidelity, zero-shot editing, which in this rebuttal can also be migrated to other data domains easily. Fig.2 and Fig.5 demonstrate our optimization pipeline can be easily applied to other backbones like PanoHead and EVA3D.
>
> Finally, we would like to emphasize that our contribution is not only EG3D-based photorealistic semantics-driven editing but also relighting. We believe that our optimization pipeline starts a new approach in semantics-driven editing and relighting given any NeRF GAN, and thus has a good impact and can spawn worthwhile future work in the years to come.

---

> > ### Comment · Reviewer_4TUt · 2023-08-16
> > **Replying to Rebuttal by Authors**
> >
> > Thank you for addressing my concerns.
> >
> > It's encouraging to see the authors extend this work to encompass 360 human heads, based on PanoHead. However, during the current rebuttal phase, evaluating the performance on 360 heads is challenging. This point does not affect my scoring of this manuscript.
> >
> > Thank you for clarifying the PTI inversion method in Fig. 2. I no longer have any concerns regarding this particular inversion method.
> >
> > While the ID loss and VGG loss may have been introduced to counter the over-saturation problem of SDS loss, it appears that they are only attempting to find a trade-off, not a solution. The final edited results still exhibit issues with identity and skin color modifications, leading to unsatisfactory outcomes. For example, as shown in Fig. 3, skin color alters after adding a beard or glasses, and the eyes change following hair style edits.
> >
> > It's unclear why the CLIP loss would induce low image quality and apparent artifacts, while the diffusion loss can prevent these issues. Furthermore, the explanation is lacking for why cosine similarity does not account for how the current update step affects the image. There is no doubt that the cosine similarity influences the networks, but this point needs further elaboration.
> >
> > Although adding relighting to EG3D-based editing is indeed a contribution, this application seems incremental for a top-tier conference. No new techniques are evident in the illumination process, other than the introduction of an Illumination Loss from reference [60].

---

> > > ### Author Response · Authors · 2023-08-18
> > >
> > > Thank you for acknowledging the contribution of our relighting on GAN-NeRF backbone, such as EG3D and Panohead. As we have already demonstrated, our method could also be migrated to other generators to achieve 360 degrees editing. We would appreciate it if you could give us some explicit suggestions about the evaluation needed and we will include them in our paper.
> > >
> > > In addition, our optimization framework that utilizes multi-loss functions to edit NeRF in the latent space in a more 3D-consistent manner is indeed a profound contribution. This approach enhances the overall 3D consistency of the editing process and further improves the editability of GAN-NeRF based methods. Also, we show human body generation and editing by replacing EG3D with EVA3D, as shown in Fig.5 in our rebuttal material. We believe our optimization pipeline starts a new approach in semantics-driven editing and relighting given any NeRF GAN, and thus has a good impact and can spawn worthwhile future work in the years to come.
> > >
> > > We do acknowledge the significance of disentangled editing and we have evaluated and shown our advantage. However, we would like to mention that semantic editing without changing the original Identity is a trade-off and long-lasting problem. For editing such as age and gender, such a trade-off is inevitable. Many attempts have been made to address this problem. The current methods for disentangling the latent space such as StyleSpace[1] and InterFaceGAN[2] rely on the pre-trained regressors or classifiers to identify style channels in S space or the hyperplanes. The S space method proposed by StyleSpace improved the Disentanglement criteria from 0.54 to 0.63 according to Table 1 in the StyleSpace paper. (1 means full disentanglement). On the contrary, our method does not require regressors or classifiers. This means that our method works for any attribute described by languages. Furthermore, Fig.3 and Tab.2 demonstrate our method achieves better disentanglement compared to InterFaceGAN, as the latent space for 3D GANs is more disentangled than that of 2D GANs.
> > >
> > > Here we clarify our explanation about why SDS guidance outperforms CLIP guidance by providing a more comprehensive explanation. First, During the optimization process using CLIP guidance, the optimization is based on the current local gradient thus it may overshoot and lead to degraded images. In the SDS setting, just as the training process of diffusion model, time t is sampled randomly to minimize the expectation with respect to t, given the text prompt. The t sampled will determine the weight of diffusion guidance, which affects the magnitude of the gradient. This offers a more robust updating process.
> > >
> > > In addition, diffusion models focus on high-quality generation, while CLIP focuses on learning cross-modal representations for understanding the relationship between images and text. However, as discussed in [3], CLIP and many other multi-modal understanding models suffer from ‘bag-of-objects’ kind of representations, possibly due to the contrastive pre-training settings. They perform well understanding nouns, but perform badly in understanding visual language concepts beyond object nouns, like attributes and relations. In our context, the input text prompt involves many non-object concepts, like emotions, colors, and other attributes of faces and decorations (e.g., “blue” hair, a “bald” man, an “Eastern” face, a “manly” woman, an “angry” lady, a woman “wearing” a pair of glasses, etc.). When using CLIP guidance, the rendered image is encoded to calculate cosine similarity with the text embedding, during which spatial and attributes information in the rendered image are lost, with focus on nouns. Thus, CLIP guidance easily “losses control” over the rendered images and generate degraded images, due to difficulty in understanding non-object concepts. In contrast, noise is injected during the training of diffusion models, which allows the denoiser to incorporate more spatial information during the iteration process. More concepts beyond object nouns are also considered by diffusion models. Therefore, we believe that SDS guidance provides more strict “control” and thus higher quality guidance.
> > >
> > > In short, we adopt SDS guidance due to its: (1) Stronger ability to alleviate trapping at a local minimum; (2) Better comprehension of concepts beyond object nouns; (3) Better consideration of spatial information.
> > >
> > > [1] Zongze Wu, et al. “StyleSpace analysis: Disentangled controls for StyleGAN image generation.” arXiv:2011.12799, 2020.
> > >
> > > [2] Yujun Shen, et al. “InterFaceGAN: interpreting the disentangled face representation learned by GANs.” arXiv preprint arXiv:2005.0963
> > >
> > > [3] Paola Cascante-Bonilla, et al. “Going beyond nouns with vision language models using synthetic data.” arXiv preprint arXiv:2303.17590.

---

> > > > ### Comment · Reviewer_4TUt · 2023-08-21
> > > >
> > > > The author has addressed some of my concerns, including the usage of SDS loss in comparison to the CLIP constraint. However, I still have reservations regarding identity preservation and disentanglement.
> > > >
> > > > (1) The final edited outcomes continue to display problems with identity and skin color alterations when expressions are changed (e.g., in Fig.1, the addition of glasses leads to changes in the ears, mouth, and eyebrows; in Fig.3, the man becomes unrecognizable from the previous input upon the addition of a mustache). The authors have acknowledged this issue, describing it as a trade-off and a long-standing problem. I hope that they can provide evidence to substantiate this claim.
> > > >
> > > > (2) The StyleSpace (S space) demonstrates superior disentanglement capabilities compared to both W space and W+ space. I'm curious to know why the authors do not conduct semantic editing using SDS loss in the S space, as was done in [Ref1].
> > > >
> > > > [Ref1] Wang, Can, et al. 'Cross-domain and disentangled face manipulation with 3D guidance.' IEEE Transactions on Visualization and Computer Graphics 29.4 (2022): 2053-2066."

---

> > > > > ### Author Response · Authors · 2023-08-21
> > > > >
> > > > > Thank you for giving us some suggestions.
> > > > >
> > > > >  (1) The paper [1] presents the main contribution of discussing the tradeoffs between distortion, perception, and editability. Additionally, it demonstrates that the distortion-perception tradeoffs exist not only between $W$ and $W_k^*$, but also within the  $W_k^*$ space itself. The trade-off problems are also mentioned in [2] and [3].
> > > > >
> > > > > (2) According to section 7 of the StyleSpace paper [4], Latent optimization in $W+$ and $S$ spaces has more flexibility than in $W$, enabling a closer reconstruction of the input image. Figure 18 of StyleSpace paper demonstrates that the visual accuracy of the reconstruction is the highest when optimizing in $S$, followed by $W+$, and is the lowest for $W$. However, the extra flexibility may result in latent codes that do not lie on the generated image manifold, and attempting to manipulate such codes typically results in unnatural artifacts. Thus, conversely to reconstruction accuracy, the manipulation naturalness is best when the latent optimization is done in $W$, followed by $W+$, and the worst for $S$ (see Figure 19 of StyleSpace paper). Thus, to make a compromise of the tradeoffs between distortion, perception, and editability, we choose the $W+$ space.
> > > > >
> > > > > [1]: Omer Tov, Yuval Alaluf, Yotam Nitzan, Or Patashnik, and Daniel Cohen-Or. Designing an encoder for stylegan image manipulation. ACM TOG, 40:1–14, 2021
> > > > >
> > > > > [2]: Yuval Alaluf, Or Patashnik, and Daniel Cohen-Or. Restyle: A residual-based stylegan encoder via iterative refinement. In ICCV, 2021.
> > > > >
> > > > > [3]: Peihao Zhu, Rameen Abdal, Yipeng Qin, John Femiani, and Peter Wonka. Improved stylegan embedding: Where are the good latents? arXiv preprint arXiv:2012.09036, 2020.
> > > > >
> > > > > [4]: Zongze Wu, Dani Lischinski, and Eli Shechtman. Stylespace analysis: Disentangled controls for stylegan image generation. In CVPR, 2021.

---

### Official Review · Reviewer_1wAz · 2023-07-06

**Soundness:** 3 good
**Presentation:** 3 good
**Contribution:** 2 fair
**Rating:** 5
**Confidence:** 4

**Summary:**

The authors proposes an approach for EG3D-based image editing in this paper.

To be specific, they utilize modified Score Distillation Sampling (SDS) on a pretrained EG3D along with reconstruction and ID loss for semantic editing (e.g., change age, eyeglasses, and hair) and relighting from a single image.

The results show good editing fidelity and 3D geometry consistency.

**Strengths:**

The paper shows some strengths:
- The results from a single image show fair identity preservation, editing fidelity and 3D geometry consistency.
- The proposed method can be migrated to other data domain, e.g., cats and cars.


**Weaknesses:**

There are some weaknesses.
- Inference time. The proposed method is an optimization-based method, which means this usually requires long time. I'd like to see what the inference time is for FDNeRF and how it compares with other methods, e.g., StyleCLIP, InterfaceGAN, and DPR for relighting.
- Missing citations and limited novelty. Using SDS score for face editing is not new. For example, [1] uses the SDS score for Style transfer. [2] considers the SDS score as an alternative for multiple applications.
- Missing experimental results. For face editing, it is common to use the ID similarity before and after editing to evaluate the ID preservation [3]. The paper does not include ID similarity as a metric to evaluate it.


[1] Song, Kunpeng, et al. "Diffusion guided domain adaptation of image generators." arXiv preprint arXiv:2212.04473 (2022).

[2] Cohen-Bar, Dana, et al. "Set-the-Scene: Global-Local Training for Generating Controllable NeRF Scenes." arXiv preprint arXiv:2303.13450 (2023).

[3] Roich, Daniel, et al. "Pivotal tuning for latent-based editing of real images." ACM Transactions on graphics (TOG) 42.1 (2022): 1-13.

**Questions:**

- A minor suggestion for "Seam" effect of the face. For example, in Figure 3 "An old man" case, we can see the "seam" artifact along the edge of the faces. This is probably because of the imbalanced camera pose distribution of FFHQ.  A possible solution is proposed by IDE3D [4] by using a density regularization.

[4] Sun, Jingxiang, et al. "Ide-3d: Interactive disentangled editing for high-resolution 3d-aware portrait synthesis." ACM Transactions on Graphics (ToG) 41.6 (2022): 1-10.

**Limitations:**

The paper has already discussed possible ethical issues brought by the technique in the paper.

---

> ### Author Rebuttal · Authors · 2023-08-10
>
>
> We thank you for your thoughtful reviews and suggestions. We are encouraged that you appreciate the two strengths of our work.
>
> Following your suggestions, we conduct more comparisons and evaluate them quantitatively using ID similarity. The results are given in the pdf file as shown in Fig.1 and Fig.3. For DPR relighting comparison, in the main paper, we render a video with fixed lighting conditions. As a 2D method, DPR fails to preserve consistent lighting when the image is conditioned from different viewpoints. We find that this is quite obvious and do not include these sanity tests in the rebuttal to save space. We will put more comparisons in the supplementary upon publication.
>
> The inference time for our method is 12 mins (NVIDIA GTX 1080 GPU), compared with 10-12 hour training of the Latent Mapper of StyleCLIP. The inference time of InterfaceGAN is fast (inversion then add latent code with normal vector), yet the quality of editing is limited by the classifier and latent space of GAN (whether it is inherently entangled). The acquisition of both a classifier and the latent boundaries takes hours of training and labeled data (for the classifier).
>
> We acknowledge the missing citations and we will cite the mentioned papers. Our method differs from "Diffusion guided domain adaptation of image generators'' which focuses on adapting the generator to new domains, while our method focuses on finding the suitable latent code given the generator. As for "Set-the-Scene,'' we consider the work as a concurrent work to ours and our tasks are very different. We thank your suggestion of PTI and utilize it in our inversion process of Panohead. We also thank your suggestion on the "seam effects.''
>
> Please see the sections in the responses to common concerns for our novelty and motivation.
>
> Finally, we would like to emphasize that our contribution is not only EG3D-based photorealistic semantics-driven editing but also relighting. We believe our optimization pipeline starts a new approach in semantics-driven editing and relighting given any NeRF GAN, and thus has a good impact and can spawn worthwhile future work in the years to come.

---

> > ### Comment · Reviewer_1wAz · 2023-08-14
> >
> > Thanks for answering my questions. I'm glad to see that more qualitative (Fig.1 and Fig.3) and quantitative (Table 2) results in the rebuttal.
> >
> > For the inference time, I think it it not fair to compare with StyleCLIP's latent mapper's training time. Although it takes 10-12 hours to train a mapper, it is much faster for it to do an inference pass. Also for InterfaceGAN, it is indeed that preparing training data and labels takes more time, but its inference time is also fast. It is probably more proper to highlight both the training time and the inference time for a fair comparison in the revision.
> >
> > For the novelty and motivation, using SDS for the latent codes has not been discussed in the prior work before. It is still valuable to check how SDS works in the latent space. However, I'm confused about your answer to the difference between SDS guidance and CLIP guidance. Especially "Second, the CLIP loss relies on cosine similarity, which lacks knowledge of how the current updated step will affect the image...On the contrary, the reverse process (denoising) of SDS tries to minimize the score of the current image conditioned on the text prompt at all iterations, which provides more consistent guidance...", in this context, do you refer to that using CLIP loss has no control over all the iterations during the optimization?
> >
> > I'd like to raise the rating if the follow-up questions are well answered. Thanks for your efforts!

---

> > > ### Author Response · Authors · 2023-08-16
> > >
> > > Thank you very much for acknowledging our technical contribution, and your advice on training and inference time comparison. We will qualify the comparison with StyleClip and InterfaceGAN, by properly highlighting both the training time and the inference time for fair comparison in the revision as you suggested.
> > >
> > > Here we clarify our explanation about why SDS guidance outperforms CLIP guidance by providing a more comprehensive explanation. First, During the optimization process using CLIP guidance, the optimization is based on the current local gradient thus it may overshoot and lead to degraded images. In the SDS setting, just as the training process of diffusion model, time t is sampled randomly to minimize the expectation with respect to t, given the text prompt. The t sampled will determine the weight of diffusion guidance, which affects the magnitude of the gradient. This offers a more robust updating process.
> > >
> > > In addition, diffusion models focus on high-quality generation, while CLIP focuses on learning cross-modal representations for understanding the relationship between images and text. However, as discussed in [5], CLIP and many other multi-modal understanding models suffer from ‘bag-of-objects’ kind of representations, possibly due to the contrastive pre-training settings. They perform well understanding nouns, but perform badly in understanding visual language concepts beyond object nouns, like attributes and relations. In our context, the input text prompt involves many non-object concepts, like emotions, colors, and other attributes of faces and decorations (e.g., “blue” hair, a “bald” man, an “Eastern” face, a “happy” lady, a woman “wearing” a pair of glasses, etc.). When using CLIP guidance, the rendered image is encoded to calculate cosine similarity with the text embedding, during which spatial and attributes information in the rendered image are lost, with focus on nouns. Thus, CLIP guidance easily “loses control” over the rendered images and generates degraded images, due to difficulty in understanding non-object concepts. In contrast, noise is injected during the training of diffusion models, which allows the denoiser to incorporate more spatial information during the iteration process. More concepts beyond object nouns are also considered by diffusion models. Therefore, we believe that SDS guidance provides better “control” and thus higher quality guidance.
> > >
> > > In short, we adopt SDS guidance due to its: (1) Stronger ability to alleviate trapping at a local minimum; (2) Better comprehension of concepts beyond object nouns; (3) Better consideration of spatial information.
> > >
> > > [5] Paola Cascante-Bonilla, et al. “Going beyond nouns with vision language models using synthetic data.” arXiv preprint arXiv:2303.17590.

---

> > > > ### Comment · Reviewer_1wAz · 2023-08-17
> > > >
> > > > Thanks for the explanations! I'd like to raise my score from 4 to 5.

---

### Official Review · Reviewer_M2ze · 2023-07-06

**Soundness:** 3 good
**Presentation:** 3 good
**Contribution:** 3 good
**Rating:** 5
**Confidence:** 4

**Summary:**

This article primarily introduces a method called Face Diffusion NeRF (FDNeRF) for 3D face reconstruction and editing. The key feature is the use of pre-trained 3D GAN and 2D latent diffusion models as priors, allowing users to operate and construct Face NeRF without requiring explicit 3D data. FDNeRF also incorporates carefully designed lighting and identity preservation losses, as well as multimodal pre-training, enabling users to have some control during the editing process. Facial NeRF can be created and edited using only single-view images, text prompts, and explicit target lighting. The article also compares FDNeRF with existing editing methods, evaluating its editability, editing effects, flexibility, and rendering quality through experimental assessments.


**Strengths:**

FDNeRF utilizes the EG3D network to reconstruct high-quality and realistic 3D facial models from a single image.
FDNeRF adopts a stable diffusion model as a guiding model, allowing effective semantically-driven editing of the 3D model based on input textual prompts.
FDNeRF introduces a novel lighting loss function that enables explicit control of the lighting of the 3D model in a view-consistent manner.
Compared to other methods, FDNeRF achieves higher computational efficiency and can generate high-quality 3D models in a shorter time frame.

**Weaknesses:**

FDNeRF primarily focuses on facial reconstruction and utilizes the EG3D network to achieve high-quality 3D reconstruction. This specific network architecture may not perform as effectively for other objects or scenes.
While FDNeRF allows for semantically-driven editing, the level of control over specific facial features or details may be limited. Fine-grained editing capabilities, such as manipulating individual facial landmarks or expressions, may not be as precise or controllable as Fig.3.
As with any AI-based system, FDNeRF may be susceptible to biases and inaccuracies. The training data and algorithms used in FDNeRF can inadvertently introduce biases or produce inaccurate results, particularly when dealing with diverse or underrepresented populations.


**Questions:**

1. What efforts did this work make to ensure identity consistency in cross-domain image editing? Is it still heavily constrained by the field of face reconstruction and editing?
2. From the results of the article, it seems that lighting remains faithful to facial changes regardless of pose limitations. Is this reasonable? Should lighting be fixed in the environment?


**Limitations:**

The editing effect of FDNeRF is influenced by semantic ambiguities in the input text prompts. It is also limited by GAN, which restricts the controllable range of poses.

---

> ### Author Rebuttal · Authors · 2023-08-10
>
> We thank you for your thoughtful reviews and suggestions.
>
> For your concerns about editing beyond the photorealistic human face domain, our diffusion-guided NeRF generator latent vector optimization pipeline can generalize to more domains by replacing EG3D with other NeRF generators. Fig.2 demonstrates the results of replacing the EG3D with the PanoHead which extends EG3D from frontal views to full head views in 360$^\circ$. Our method still can achieve high-fidelity semantic editing. Fig.5 presents editing examples of articulated human bodies, a very challenging domain. We use EVA3D (ICLR2023 Spotlight), a SOTA human body NeRF generator to replace EG3D while deleting the identity loss, which works only on human faces.
>
> For feature loss, given a human body image, we either estimate the camera pose and calculate feature loss with a rendered image from the same viewpoint in each iteration, or perform inversion to find the corresponding latent vector of the input image and compute the feature loss using single or multiple pairs of images rendered from random view directions. While EVA3D authors claimed both high-quality inversion and camera pose estimation can be achieved in the paper, unfortunately, at the time of writing this rebuttal the official codes have not been released.
> Therefore, for experimental purposes, we perform editing on images generated from EVA3D so that the camera is known.
>
>
> During each iteration, we compute the feature loss using the input image and a rendering of the current human body NeRF from the same viewpoint. Diffusion loss and the optimization pipeline are the same as our EG3D version. As shown in Fig.5, despite the suboptimal generation quality of EVA3D, our editing is still flexible and reasonable. We apologize that we do not have time to implement EVA3D inversion and camera pose estimation. We will release code for more domains after publication.
>
> Regarding the fine-grained editing capabilities, we acknowledge its advantage of precise control. However, relying only on predefined controlling parameters is suboptimal as semantically complex attributes such as age, gender, and makeup are hard to define (as in the case of FeNeRF and IDE3D). For the purpose of fine-grained control, a natural extension of our method could replace the origin Stable Diffusion by ControlNet, the generation process of which could be controlled by landmarks.
>
> For the lighting formulation, we could define a lighting position for the image. Then when viewing the image from different directions, the lighting source should be fixed (Imagine a 3D image captured, its lighting direction is fixed). Also, it is easy to change the lighting direction in our method explicitly as shown in the paper.

---

> > ### Comment · Reviewer_M2ze · 2023-08-21
> >
> > Thank you for your rebuttal. I have read it and all my questions and concerns have been addressed.
> >
> >  I will respond with points for further discussion.

---

### Official Review · Reviewer_JGgm · 2023-07-06

**Soundness:** 3 good
**Presentation:** 3 good
**Contribution:** 3 good
**Rating:** 6
**Confidence:** 4

**Summary:**

This work proposes a 3D generative model-based optimization pipeline for 3D editing from a single image which is a useful and challenging task. The proposed pipeline achieves text and lighting-driven editing by reusing several existing pre-trained models. By carefully designing the losses trade-off, it achieves reasonable results while keeping 3D view consistency.

**Strengths:**

- The proposed pipeline for semantic-driven NeRF 3D editing from a single image is composed of several pre-trained models such as EG3D, StableDiffusion, ArcFace, facial SH lighting estimator, and VGG. The editing process is formulated as an optimization problem with several objectives to trade-off the identity preservation, image quality, and editing text/view/lighting consistency. The pipeline achieves reasonable results while keeping 3D view consistency by reusing existing prior knowledge.

- Several engineering modifications have been done to make the whole pipeline work, including differential operation replacement, latent regularization, face view choices. Although the whole system is a bit heavy and the technical novelty is moderate, it can be applied to real photo via GAN inversion and achieves reasonable semantic-driven NeRF 3D editing results with control from text and lighting.

- The supplementary video demonstrates the effectiveness of the proposed method in semantic editing while preserving the identity/view consistency. The paper organization and illustrations are easy to follow.

**Weaknesses:**

- The comparisons with prior works (e.g. InterfaceGAN, FENeRF, StyleClip) are not enough. There are only a few samples illustrated in Fig. 4. It would be good to see more comparisons and discussions. As claimed that FDNeRF is not limited to human faces, it would be good to show more results on other datasets, especially edited results.

- It is mentioned in line 145 “VGG16 image encoder and ArcFace … due to mismatched viewing directions”. It would be interesting to see if small view changes can benefit the performance since VGG features and ArcFace features are not that pixel-aligned, especially for ArcFace features.

- Clearly boundary can be observed between face and hair, e.g., the edited views of the last row in Figure 3. More discussion and analysis on this issue would be appreciated.

**Questions:**

More comparisons and discussions as mentioned in the weakness are required.

**Limitations:**

The potential negative societal impact should be discussed.

---

> ### Author Rebuttal · Authors · 2023-08-10
>
> We thank you for your thoughtful reviews and suggestions. We are encouraged that you appreciate the three strengths of our work. Following your suggestions, we compared our work with InferfaceGAN and StyleClip quantitatively and qualitatively in Fig.1, Fig.3, Tab.1, and Tab.2. To show the generalizability of our model, we replaced EG3D with PanoHead and EVA3D respectively.
>
> Fig.2 demonstrates the results of replacing EG3D with PanoHead which extends EG3D from frontal views to full head views in 360$^\circ$. Our method can still achieve high-fidelity semantic editing. Since the structure of PanoHead is almost the same as EG3D, structure, loss terms, and other details of PanoHead-based pipeline are almost the same as the EG3D-based pipeline.
>
> Fig.5 presents editing examples of articulated human bodies, a very challenging domain. We use EVA3D (ICLR2023 Spotlight), a SOTA human body NeRF generator to replace EG3D while deleting identity loss, which works only on human faces.
>
> For feature loss, given a human body image, we either estimate the camera pose and compute the feature loss with a rendered image from the same viewpoint in each iteration, or perform inversion to find the corresponding latent vector of the input image, and then compute the feature loss using single or multiple pairs of images rendered from random view directions. While EVA3D authors claimed both high-quality inversion and camera pose estimation can be achieved in the paper, unfortunately, at the time of writing this rebuttal the official codes have not been released. Therefore, for experimental purposes, we perform editing on images generated from EVA3D so that the camera is known.
>
> During each iteration, we compute the feature loss using the input image and a rendering of the current human body NeRF from the same viewpoint. Diffusion loss and the optimization pipeline are the same as our EG3D version. As shown in Fig.5, despite the suboptimal generation quality of EVA3D, our editing is still flexible and reasonable. We apologize that we do not have time to implement EVA3D inversion and camera pose estimation. We will release code for more domains after publication.
>
> EG3D contains the geometry constraint of the human head. Thus, if the identity is preserved in one view, images rendered from other views should also preserve the same identity. We did some experiments which show that there is no consistent improvement of adding more constraints on the images rendered from small view changes. Due to the space limit, we do not show the results of the experiments and will add them in our camera-ready paper.

---

> > ### Comment · Reviewer_JGgm · 2023-08-20
> >
> > Thanks for the rebuttal. I have read other reviews and authors' feedback. The rebuttal has addressed all my concerns. Several recent new backbones have been adopted in the rebuttal and demonstrate better results.  Please also integrate these results in the final version and code release. I would keep my initial rating.

---

### Author Rebuttal · Authors · 2023-08-10

We thank all reviewers for their insightful comments and appreciation of our work. Below we address the issues. The referred figures and tables are in a separate PDF file. Following the given review sequence, we use R1, R2, R3, R4 to denote reviewer JGgm, M2ze, 1wAz, 4TUt respectively.


### [R1, R3, R4] More Comparisons and Evaluations.
We conduct both qualitative and quantitative comparisons with InterfaceGAN and StyleCLIP.


In Fig.1, since StyleClip proposed two methods Latent Optimization and Latent Mapper, we compared them separately. We adopt the same text example as Fig.4 in the StyleClip paper. For clarity, we only present the results of our method and the Latent Optimization method. Please refer to the original paper for the results of the Latent Mapper.

The qualitative results demonstrate that our method outperforms the Latent Optimization method in terms of consistency with the text prompt and identity preservation, which are also corroborated in the quantitative results in Tab.1. It is important to note that the Clip loss alone cannot fully capture the visual perception of distances between images and the semantic meanings of text prompts. In some cases (e.g., Fig.1, 1st-row 3rd-column, 3rd-row 3rd-column, and 4th-row 5th-column), the results of the Latent Optimization method exhibit degradation while the Latent Optimization method achieves the lowest Clip loss. On the other hand, our method consistently produces reasonable results across all cases. Even when compared to the latent mapper method, which requires 10--12 hours of training for a specific text input, our method achieves a comparable level of visual quality. Our method only needs around 10mins inference time and no training is needed for a specific text prompt. (Times are measured on a single NVIDIA GTX 1080ti GPU.)

 As shown in Fig.3, we compare with InterfaceGAN on more attributes. Note that as the latent direction found by the classifier could be entangled, the editing results of InterfaceGAN often exhibit undesired changes in other attributes which leads to identity changes. Tab.2 quantitatively verifies that our method preserves Identity better while achieving the target editing.

### [R2, R4] Extended Head Pose Range.
The range of head pose is limited by the trained EG3D generator as most facial images are in front view. Recent Panohead addresses this problem and enables head generation in 360 degrees. Our method can employ Panohead by replacing the generator, as shown in Fig.2.


### [R3, R4] SDS in Latent Space \& Difference with CLIP
We acknowledge the usage of SDS in other papers. On the other hand, our utilization of SDS loss on the latent space of trained GAN is motivated by high-fidelity 3D editing with language guidance. As shown in Fig.4, the original SDS can only generate a blurry human head, where the process can collapse with less careful prompts. This is because of the over-saturation and low-diversity problems of SDS. There are methods to directly apply diffusion in 3D human heads (Rodin), yet they rely on 3D ground truth data (synthesized), which limits the quality and diversity of 3D head diffusion models, while GAN requires no specific 3D data and achieves high-quality 3D results. Therefore, with trained GAN as a constraint, our SDS can effectively utilize the latent space.


During our comparison with StyleCLIP, we empirically found that the CLIP loss tends to cause degraded images (i.e., images with low image quality and obvious artifacts, not similar to the original image, not coherent to the text prompts). We suspect several reasons for this phenomenon. First, the denoising process of SDS adds noise at each iteration, which reduces the possibility of being trapped in the local minimum. Second, the CLIP loss relies on cosine similarity, which lacks knowledge of how the current updated step will affect the image. Thus these possible causes can deviate from the original image drastically. On the contrary, the reverse process (denoising) of SDS tries to minimize the score of the current image conditioned on the text prompt at all iterations, which provides more consistent guidance.

### [R1, R3] Artifacts Near the Edge
We also observe such artifacts and we thank the reviewer for their suggestions. This could also be caused by the relevant face preprocessing and training camera pose of EG3D. When shifted to Panohead, such artifacts thankfully no longer appear.

### [R1, R2] Cross Domain Editing & Generation
We extend our method to the articulated human body domain, as shown in Fig.5, where we replace EG3D in the submission by EVA3D, a human body NeRF generator. The excellent results we obtained during the short NeurIPS rebuttal period indicate that our pipeline can be generalized to various domains and different GAN architectures.


### [R4] Inversion Ablation
Following your suggestions, we implemented a PTI version of our method. The code of the PTI version will also be released. The results presented in Fig.2 have already incorporated the PTI fine-tuning.

---

### Decision · Program_Chairs · 2023-09-21

**Decision:**

Accept (poster)

**Comment:**

The paper received mixed reviews initially, which remained still mixed even after discussions. The remaining concern by Reviewer 4TUt after the rebuttal was that there are still issues with the quality of the results regarding identity preservation and disentanglement. While the results are imperfect, it is clear that the proposed method is still an improvement over the compared baselines. Given that the majority of the reviewers are convinced, and that the remaining issue does not seem critical, the AC recommends accepting the paper.